# Inequalities in the identification and management of common mental disorders in the perinatal period: An equity focused re-analysis of a systematic review

**Stephanie L. Prady** [1]*, **Charlotte Endacott**[2], **Josie Dickerson**[2], **Tracey J. Bywater**[1], **Sarah L. Blower**[1]

1 Department of Health Sciences, University of York, York, United Kingdom, 2 Bradford Institute for Health Research, Bradford Royal Infirmary, Bradford, United Kingdom

* stephanie.prady@york.ac.uk

**Data Availability Statement:** All relevant data are within the manuscript and its Supporting Information files.

## Abstract

### Objective

Maternal mental health problems in the perinatal period can cause significant distress and loss of functioning, and can have lasting impact on children. People living in disadvantage are at risk of health inequalities, including for perinatal mental health. A review of current guidance found that overall implementation of the UK detection and management strategy was satisfactory, but equity was not considered in the review. Greater understanding of implementation equity is needed. We aimed to reanalyse an existing systematic review on the implementation of current guidance for the identification and management of perinatal mental health problems for equity.

### Methods

Studies reporting the presence or absence of variation by a social, economic or demographic group were quality appraised and the presence and direction of disparity tabled. We calculated standardised absolute prevalence estimates for overall detection and management, and absolute and relative estimates by determinants grouping. A thematic analysis of the studies that examined potential reasons for disparity was undertaken.

### Results

Six studies, with no major quality concerns, provided consistent evidence of reduced identification and management for ethnic minority women, both those who do, and do not, speak English. There was less consistent evidence of inequality for other axes of social disparity and for characteristics such as age, parity and partnership status. Explanations centred on difficulties that translation and interpretation added to communication, and hesitancy related to uncertainty from healthcare providers over cultural understanding of mental health problems.

**Funding:** This report is independent research funded by the Medical Research Council award MC_PC_17210 (SLP). SLP, SB, TB and JD are also supported by the National Institute for Health Research Yorkshire and Humber ARC (NIHR200166). The views expressed in this publication are those of the author(s) and not necessarily those of the National Institute for Health Research or the Department of Health and Social Care. The funders had no role in study design, data collection and analysis, decision to publish, or preparation of the manuscript.

**Competing interests:** The authors have declared that no competing interests exist.

## Conclusion

The identification and management of perinatal mental health problems is likely to be inequitable for ethnic minority women. Further systems-based research should focus on clarifying whether other groups of women are at risk for inequalities, understand how mismatches in perception are generated, and design effective strategies for remediation. Inequalities should be considered when reviewing evidence that underpins service planning and policy decision-making.

## Introduction

Common mental disorders (CMD) such as anxiety and depression affect around one in four people a year [1] with a similar rate of occurrence in women during pregnancy [2]. CMD during the perinatal period (pregnancy and 1 year after the birth) can cause significant distress and loss of functioning by interfering with biological, attachment and parenting processes. For some, this disruption can track through to the children causing lifelong impacts [3, 4]. The cost per case to society for perinatal mental health problems in 2014 was estimated at around £74,000 for depression, and £35,000 for anxiety [5].

CMD in the community are more likely to occur in people who are demographically, socially or economically disadvantaged, through processes such as stress and discrimination, e.g. [6–10]. This is health inequality; unfair and avoidable differences in health caused by unequal social conditions [11]. In this paper we use inequality as synonymous with inequity and disparity [11]. People with anxiety and depression who are disadvantaged may be less likely to have their disorder recognised in the healthcare system, be offered, and uptake, treatment [12–14]. This perpetuates the intertwining of poor health with disadvantage and causes significant health inequality. The UK National Health Service (NHS) is bound to uphold the Equality Act 2010, which grants the rights of people with protected characteristics from unfair treatment and discrimination [15].

Systematic population (universal) screening is the identification of those at risk of a health condition [16]. In the UK, the National Screening Committee (NSC) is the body responsible for providing advice about whether screening for a particular condition should be applied. At the last evidence review in 2019, the NSC upheld their previous recommendation against universal screening for the identification of perinatal CMD [17]. The evidence review found that while the condition and its negative consequences are well understood, there are evidence gaps in the performance of screening tests for anxiety and effective interventions for screen-detected women [18].

In the absence of a screening programme, the current National Institute for Health and Care Excellence (NICE) guidance lays out how healthcare professionals in the UK should identify maternal perinatal CMD [19]. The current strategy is for all people involved with a mothers care to "...consider asking... [brief] identification questions as part of a general discussion about a woman's mental health and wellbeing..." [19, item 1.5.4]. The Whooley questionnaire and GAD-2 are two-item screening tools designed to indicate the presence of anxiety (GAD-2) and depression (Whooley) symptoms [20, 21]. If positive, the pathway indicates further investigation, treatment or referral as necessary, and the use of standardised measures to confirm severity. The NSC review examined the evidence for implementation of this current strategy (Question 6—Is clinical detection and management currently well implemented in the UK?) and found that 'most' women are asked about their mental health [18].Variation in

implementation by socio-economic group was not assessed, indeed inequalities was not examined in of the research questions.

Perinatal women come into contact with healthcare professionals in multiple services, including midwives (MWs), health visitors (HVs) and general practitioners (GPs). In the UK this provision is collectively known as universal services, and are designed to meet general expected needs, including assessment and some management of mental health problems as indicated in the NICE guidance. Previous work has indicated that there may be inequalities in the detection and management of maternal perinatal CMD in general practice [22, 23] but it is unclear whether this extends to other providers of universal services and the evidence base has not been reviewed.

We aimed to examine inequalities in the current implementation of the NICE guidance by reanalysing the studies included under Criterion 15—Implementation of the current strategy and Question 6 of the NSC review "Is clinical detection and management currently well implemented in the UK?" [17, 18]. We defined disadvantaged groups using the protected characteristics described by the PROGRESS acronym [24] and added age, parity and marital / relationship status as additional contextual disadvantage characteristics for this population.

Our research questions were (1) is implementation of current UK guidance related to detection and management equitable? and, (2) if implementation is not equitable, what are the potential reasons for this?

## Methods

We reanalysed studies included in an existing systematic review (18), following the equity extension for the Preferred Reporting Items for Systematic Reviews and Meta-Analysis (PRISMA-E) guidance [25]. We did not register the protocol.

Eligible studies were UK studies that were included under Criterion 15—Implementation of the current strategy and Question 6 of the NSC review "Is clinical detection and management currently well implemented in the UK?" [18], henceforth 'NSC implementation review'. This is a major, comprehensive and contemporary review of UK evidence that we felt would be a fair representation of the studies in which we were interested. We initially examined whether the questions' search terms, study inclusion criteria and outcomes in the NSC implementation review were suitable for our purpose. For example, if studies that *only* reported on equity considerations were excluded, then the body of evidence would not be adequate for our purpose. We concluded that the questions, terms, criteria and outcomes were suitable for our purpose.

### Search strategy and study selection

**NSC implementation review.**   The NSC implementation review electronically searched Medline, Embase, PsycINFO and the Cochrane Library. The search was conducted in February 2018 and papers published between 2011 and 2018 were considered for inclusion. Search terms included free text and subject headings with variations around the terms depression, postpartum and study types. Exclusions in the search strategy were non-English language and publication types (e.g. editorials and commentaries). The search strategy is presented in full in the original review [18]. Five primary studies from the 746 yielded in total were included in the NSC implementation review along with a meta-synthesis of qualitative research [26]. These five papers and the meta-synthesis were considered for our review.

**This review.**   The aim of the meta-synthesis by Ford et. al. [26] that was included in the NSC implementation review was not equity; thus any data on inequalities present in the primary studies were not synthesised in the meta-synthesis and hence were also not drawn out in the NSC implementation review. We therefore decided to obtain and include at the study level

the 4/5 studies conducted in the UK included in Ford et. al. [26]. During this process we identified a second report (containing quantitative data) associated with an included mixed methods study, which was not included in either Ford et. al. [26] (because it was quantitative), or the NSC implementation review. We included both reports, treating them as the same study, but acknowledge that the additional data reported is missing from NSC assessment.

## Data extraction

We piloted a data extraction form on three included papers to ensure adequacy across study designs and to check consistency between reviewers. We extracted information on the study design characteristics and participant exclusion criteria, setting and participant characteristics, perinatal timing, methods of analysis, outcomes related to identification and management, and any report (qualitative or quantitative) of identification and management outcomes by equity. Data was extracted by one author and checked by another.

## Quality appraisal

We undertook a quality appraisal using the CASP checklists for the six papers analysed in this review [27, 28]. For consistency, we used the cohort version of the CASP checklist for studies involving cross-sectional surveys. Each study was categorised as either 'no concerns' (the study has minor limitations but these do not jeopardise findings) or 'some concerns' (the study has major limitations) or 'significant concerns' (the study has significant limitations that fundamentally jeopardise the validity or reliability of the findings). The NSC implementation review also used CASP where applicable. Quality appraisals were undertaken by one of two authors and checked by the other.

## Categorisation of determinants

We applied the broad categories of the PROGRESS acronym [24] to each paper. We classified disadvantaged groups as follows: **P**lace of residence (housing/area is worse/less safe/more deprived); **R**ace/ethnicity/culture/language (not White/minority culture/does not speak English); **O**ccupation (not employed/higher precarity/lower status job/fewer hours); **G**ender/ sex (not applied); **R**eligion (not applied); **E**ducation (less education/fewer or no qualifications); **S**ocioeconomic status (lower income/less materially well-off); and **S**ocial capital (not applied). In accordance with the–Plus characteristics of PROGRESS-Plus, we added marital / relationship status (disadvantaged = not married/partnered), age and parity as additional contextual disadvantage characteristics for this population. We classified disadvantage in age as women younger than average childbearing age, as they are likely to have fewer resources, and also older women, who are at higher risk for a lifetime mental health problem. Increased parity was classified as disadvantaged. We were interested in evaluating parity separately from age due to two potential scenarios that might decrease the likelihood of mental health conversations being raised for women with existing children, particularly within home visits; (1) the presence of accumulated pregnancies previously untroubled by mental health problems, and (2) the distraction of other young children in the household. We applied these classifications to determinants as reported in each paper, but due to measurement and reporting variation, we could not standardise across papers and in qualitative papers, reference groups were indistinct.

## Analysis and synthesis

We describe the characteristics of the studies that did, and did not, present findings by equity. We present a narrative synthesis of the equity findings in the context of their quality appraisal.

To address the first research question 'Is implementation of current UK guidance related to detection and management equitable?' we tabled the presence and direction, of any association between determinants and detection or management, where reported. Where numerical data were reported, we calculated standardised absolute prevalence estimates for overall detection and management, and absolute and relative estimates by determinants. Estimates were calculated by one reviewer and checked by another. Comparative statistics were only computed where the denominator was greater than five. For these calculations, which illustrate the potential size of any disparity, we selected four determinants from PROGRESS-Plus: markers of ethnicity or language; parity and two markers of socio-economic status (SES), one area measure of deprivation and one individual measure (education). These were selected to represent varying axes of disadvantage in perinatal women. We acknowledge the presence of, but did not represent, intersectionality (the intersecting or overlapping effects of ethnicity, SES and other characteristics that contribute to social identity and affect health) [29] in these summaries. We then synthesised the findings across studies by determinants group, in relation to their risk of bias; separately presenting information pertaining to detection and management. The two studies providing quantitative data on identification presented information from different perspectives which were not possible to synthesise. Only one study presented data on management.

To address the second research question, 'If implementation is not equitable, what are the potential reasons for this?', we undertook a thematic analysis [30]. After considered reading, we determined which studies explored potential reasons for inequity in their study and extracted all data pertinent to this. One author coded the data and from these initial codes, developed themes where codes were repeated within and between the studies. The themes were refined to ensure comprehension and that they were not repeated, and were reviewed by another author.

## Results

### Included studies

We included nine studies reported across 10 publications [22, 31–39]. Some data from one study was reported across two papers [31, 32] which were separately assessed in the NSC implementation review, and as noted in the Methods, we located and included a companion report to Khan [34].

### Characteristics of studies and reporting of equity

Six out of the nine studies reported some results by equity (Table 1). Of these six studies, three were qualitative enquiries, two were analysis of quantitative data and one used mixed methods. Only one explicitly aimed to look at disparity. These six studies analysed data and viewpoints from mothers and health professionals across antenatal and postnatal care, covering both detection and management. The perspective or data from mothers, MWs and GPs were most

**Table 1. Study characteristics.**

| Study (topic), publication type | Design, (perinatal period) | Sample size | Location | Sampling (exclusions) | Sample socio-economic status | Study aims | Reported results by equity (measurement of determinants) |
|---|---|---|---|---|---|---|---|
| Williams et. al. (2016) [39] (detection), PRP | Qualitative semi-structured interviews (AN) | n = 15 MWs<br><br>n = 20 mothers | North Bristol | Mothers participating in validation study at booking appointment; MW referring women into study | Mothers purposefully sampled on age, SES, parity | Views and experiences of AN case-finding questions | Ethnicity/language/culture<br><br>Age<br><br>Partnership attendance (not defined, as mentioned by interviewees) |

*(Continued)*

**Table 1.** (Continued)

| Study (topic), publication type | Design, (perinatal period) | Sample size | Location | Sampling (exclusions) | Sample socio-economic status | Study aims | Reported results by equity (measurement of determinants) |
|---|---|---|---|---|---|---|---|
| Redshaw & Henderson (2016) [38] (detection, management), PRP | Secondary data analysis of the national maternity survey, 'Safely delivered' (AN, PN) | Mothers who answered mental health questions: Antenatal n = 4,521 Postnatal n = 4,502 | England | Random sample of women giving birth Jan 2014 (<16 years, baby died) | Respondents more advantaged, older, born in UK | Characteristics of women who are asked about mood and mental health, and offer and update of treatment | Ethnicity (SR: White, Mixed, Asian, Black, Other) |
| | | | | | | | Education (SR: Left full-time education at age <17, 17–18, 19 +, still in education) |
| | | | | | | | SES (Area deprivation quintile) |
| | | | | | | | Partnership status (SR: single, coupled) |
| | | | | | | | Parity (SR: primaparous, multiparous) |
| | | | | | | | Age (SR: 16–19, 20–24, 25–29, 30–34, 35–39, 40+) |
| McGlone et. al. (2016) [37] (detection), PRP | Qualitative semi-structured interviews (AN) | n = 8 MWs | Northwest England | MW who regularly conduct booking visits | Population served NR | Experiences using case-finding questions and how this impacts on identification | Ethnicity/language |
| | | | | | | | Partnership attendance (not defined, as mentioned by interviewees) |
| McCauley & Casson (2013) [36] (management), PRP | Qualitative, semi structured interviews (AN, PN) | n = 8 GPs | Derry City | Selected male & female GPs with relevant experience from most & least deprived areas | Derry City contains 5 of most deprived areas in NI | Experiences of using treatment guidelines and effects on empowerment | Education |
| | | | | | | | SES (not defined, as mentioned by interviewees) |
| Khan (2015) [34] and Russell & Lang (2013) [35]** (detection, management), Reports | Surveys, literature reviews, qualitative, interviews (AN, PN) | Surveys: n = 43 GPs, n = 2,093 other health professionals (HVs, MWs & others), n = 1547 mothers Interviews: n = 3 GPs. n = 4 mothers | UK wide | GPs through RCCP, Other health professional through professional bodies, women on NetMums who were self- or doctor diagnosed with perinatal mental health problems | Netmums is reportedly slightly over-representative of those in lower socio-economic groups. Populations served by included GPs and other health professionals NR | Experiences of disclosure, identification, help seeking, access to care, support and professionals' work | Place |
| | | | | | | | Ethnicity/language/culture (not defined, as mentioned by interviewees) |
| Prady et. al. (2016) [22] (detection), PRP | Linked cohort & medical records data analysis (AN, PN) | n = 8991 women | Bradford | Women enrolled in Born in Bradford (relocated, no linked records or recruitment questionnaire, severe mental illness) | Bradford is a relatively deprived city of which Born in Bradford is representative, those analysed were more likely to be advantaged and nulliparous | Describe characteristics of distress in primary care and disparities in detection | Ethnicity/language (SR ethnicity, language used for recruitment questionnaire: White British-English, Pakistani-English, Other-English, Any-not English) |
| | | | | | | | Occupation (SR employment status: currently, previously, never) |
| | | | | | | | Education (SR highest qualification or equivalent: ≥A-level, <A-level***) |
| | | | | | | | SES (all SR. 1. No. of material items lacked on the Family Resource Survey: 0, 1–2, 3–4, 5+; 2. Area deprivation quintile: most deprived, quintiles 2–5; 3. In receipt of means tested benefits: No, Yes) |
| | | | | | | | Age (SR: <20, 21–34, 35+) |
| | | | | | | | Parity (From maternity record: Nulliparous, 1, 2–3, 4+) |
| | | | | | | | Marital status (SR: Married, cohabiting, neither) |

(*Continued*)

**Table 1.** (Continued)

| Study (topic), publication type | Design, (perinatal period) | Sample size | Location | Sampling (exclusions) | Sample socio-economic status | Study aims | Reported results by equity (measurement of determinants) |
|---|---|---|---|---|---|---|---|
| Darwin et. al. (2015) [33] (detection, management), PRP | Mixed methods; clinical records review, data collection, interviews (AN, PN) | Mothers n = 191, interview subsample n = 22 | North England | Women attending booking appointment | Sample more advantaged and younger than population served | Completion and consistency of assessments, management of potential cases, experience of referrals | No |
| Chew-Graham et. al. (2009) [32]* (detection), PRP | Qualitative flexible interviews (PN) | n = 14 HVs; n = 19 GPs; n = 28 mothers | Bristol, Manchester & London | From practices participating in and mothers with depression completing PND treatment trial | Populations served included deprived, especially in Manchester; Characteristics of mothers NR | Views on disclosure of PND | No |
| Chew-Graham et al (2008) [31]* (detection, management), PRP | Qualitative flexible interviews (PN) | n = 14 HVs; n = 19 GPs | Bristol, Manchester & London | From practices participating in PND treatment trial | Populations served included deprived, especially in Manchester | Views on detection & management of PND and effect of working arrangements between HVs and GPs | No |

AN antenatal, PN postnatal, HVs health visitors, MWs midwives, GPs general practitioners, PND postnatal depression, RCGP Royal College General Practitioners, PRP peer reviewed publication, NR not reported, SR self-reported.

\* The same health professional interviews were used for both papers

\*\* The paper by Khan [34] was not included in NSC implementation review

\*\*\* A-levels are qualifications achieved after successful completion of a further 2 years' full-time school after compulsory education ends at age 16.

often represented with HVs and other health professionals represented by a single study reported across two papers. Samples were drawn from the UK, England, Northwest England, Derry City (Northern Ireland), North Bristol and Bradford. Three of the six studies reporting results by equity examined samples from disadvantaged populations, one from a relatively advantaged population and two did not report participant characteristics.

## Quality appraisal

Quality appraisal findings are in Table 2. The limitations of included studies largely related to generalisability either due to small sample sizes [36, 37], characteristics of the setting [22, 36], under-representation of disadvantaged and younger women [38] or being based in one NHS site [22, 37]. There was limited information on the design and sample presented for one study [34, 35]. The quality appraisal process did not generate significant concerns regarding methodological quality for any of the included studies.

## Equity findings

Study results by equity in PROGRESS-Plus categories are presented in Table 2.

*Question 1. Is implementation of current UK guidance related to detection and management equitable?*

**Table 2. Results by equity.**

| Study | Study analysis methods (period and group studied) | Summary of findings—effects (Q1) ▾evidence that advantaged group is favoured ▴evidence that disadvantaged group is favoured ▸ little evidence of a difference between advantaged & disadvantaged groups | Summary of findings— explanation of effects (Q2) | Quality Appraisal |
|---|---|---|---|---|
| **Detection** | | | | |
| Williams et. al. (2016) [39] | Qualitative semi-structured interviews (AN; mothers, MWs) | **Ethnicity/language/culture** ▾ MW less likely to use case finding questions if the woman has limited English (vs good English) **Age**▾ MW less likely to use case finding questions for younger women (vs older). **Partnership attendance**▾ MW less likely to use case screening questions if a partner was present. Mothers noted partner presence may limit disclosure. | **Ethnicity/language/culture** MW felt that different cultural understandings of mental health problems would get in the way, including interpretation by an interpreter. **Age** NR **Partnership attendance** NR | CASP checklist for qualitative studies. No concerns. Recruitment was via a linked study in one English city so may lack wider generalisability. Views were based on self-report and not supplemented by observation (recall and other biases may be present). |
| Redshaw & Henderson (2016) [38] | Secondary data analysis of the national maternity survey, 'Safely delivered' (AN, PN; mothers) | **AN Asked about current MH problems** **Education** ▸, **SES** ▸, **partnership status** ▸ **Age** ▾Bivariate: 40+ less likely (76% vs 30–34 82%). Multivariable: NS. ▸ other ages. **Ethnicity**▾ Bivariate: Asian less likely (76% vs White 83%). Multivariable: Asian less likely (OR 0.67; 95% CI 0.52, 0.86). ▸Other ethnicities **Parity**▾ Bivariate: less likely multiparous 81% vs nulliparous 83%. Multivariable: NS. **AN Asked about own/family history of MH problems** **Education** ▸, **partnership status** ▸ **Age**▾ Bivariate: less likely age 40+ 83% vs 30–34 86%. Multivariable: less likely age 35–39 (0.79; 0.63, 0.99) and 40+ (0.53; 0.38, 0.74). ▸ <30. **SES**▴ Bivariate: NS. Multivariable: most deprived quintile more likely (1.32; 1.00, 1.74) vs least deprived. **Ethnicity**▾ Bivariate: Asian less likely (79% vs White 83%). Multivariable: Asian less likely (0.67; 0.51, 0.88) ▸Other ethnicities **Parity**▾ Bivariate: less likely multiparous 83% vs nulliparous 86%. Multivariable: NS. **PN Asked about mental health** **Ethnicity**▾ Bivariate: Less likely Asian 80%, Black 80%, Mixed 85%, Other 70% vs White 92%. Multivariable: Less likely mixed (0.51; 0.27, 0.94), Asian (0.37; 0.28, 0.50), Black (0.43; 0.27, 0.69) and other (0.20; 0.08, 0.50). **Education**▾ Bivariate: less likely left school < age 17, and 17–19, both 88% vs left at 19+ 92%. Multivariable NS. **Age**▾ Bivariate: less likely 16–19 (80%) vs 30–34 91%. Multivariable: less likely age 16–19 (0.32; 0.18, 0.57) and 20–24 (0.59; 0.42, 0.83). **SES**▾ Bivariate: less likely most deprived quintile 84% vs least deprived 92%. Multivariable most deprived (0.70; 0.50, 0.99). **Parity**▾ Bivariate: less likely multiparous 89% vs primaparous 91%. Multivariable: less likely multiparous (0.73; 0.59, 0.91). **Partnership status**▾: Bivariate less likely single 84% versus partnered 91%. Multivariable: less likely single (0.72; 0.54, 0.96). | N/A | CASP checklist for cohort studies. Some concerns. Response rate of 47% to postal survey is good, however young women, single women, women born outside of the UK and those living in areas of higher deprivation are under-represented. |

*(Continued)*

**Table 2.** (*Continued*)

| Study | Study analysis methods (period and group studied) | Summary of findings—effects (Q1) ▼evidence that advantaged group is favoured ▲evidence that disadvantaged group is favoured ► little evidence of a difference between advantaged & disadvantaged groups | Summary of findings— explanation of effects (Q2) | Quality Appraisal |
|---|---|---|---|---|
| McGlone et. al. (2016) [37] | Qualitative semi-structured interviews (AN; MWs) | **Ethnicity/language** ▼ Detection may be compromised in consultations using an interpreter (vs not) | **Ethnicity/language** Concerns that using an interpreter seems to take longer to undertake the booking appointment, which adds time pressure during the visit. It was implied that these pressure impacts on identification. | CASP checklist for qualitative studies. No concerns. Authors acknowledge the potential impact of an academic midwife conducting the interviews. Small sample based at one NHS site which may limit generalisability to the wider UK. |
| Khan (2015) [34] and Russell & Lang (2013) [35] | Surveys, literature reviews, qualitative, interviews (AN, PN; mothers, GPs, MWs, HVs, other professionals) | **Ethnicity/language/culture**▼ Detection may be compromised in consultations using an interpreter (vs not), and consultations with ethnic minority women (vs White women) | **Ethnicity/language/culture** Almost half of non-GP health professionals felt that cultural and translation problems constituted a significant barrier to raising a mental health conversation | Khan (2015) [34] CASP checklist for qualitative studies. Some concerns. The lack of methodological information reported limits strength of confidence in conclusions. GPs self-selected from an unknown sample. and Russell & Lang (2013) [35] CASP checklist for cohort studies. Some concerns. Response rates and representativeness not discussed. All women reported they had experienced poor mental health but unclear if this was a design choice. Includes women with formal and self-diagnoses; unclear if a standard measure was used. Sample characteristics not provided. |
| Prady et. al. (2016) [22] | Linked cohort & medical records data analysis (AN, PN, GPs) | **AN Compared to women identified with CMD, those potentially unidentified were** **Ethnicity/language** ▼ For Pakistani women who used English RRR 2.66 (95% CI 2.17, 3.26), Other ethnicity (English) 2.59 (2.00, 3.35), any woman not using English 2.33 (1.74, 3.14) vs White British women **Occupation** ▲previous employment vs current for other ethnicity (not English) ► all other groups **Education** ► all groups **SES** ▼ Pakistani (English) ► all other groups **Age** ▼ <20, White British, ► all other groups **Parity** ►all groups **Marital status** ▼ Cohabiting, for White British, ► all other groups **AN Case-finding recorded** Ethnicity/language ▼<1% of other ethnicity (not English) had records of case finding vs 4.8% White British, 2.2% Pakistani (English), 2.0% Other (English) **PN The relationship between potentially missed AN CMD and increased prevalence of PN CMD** **Ethnicity/language** ▼ White British and Pakistani (English) ► other groups **PN Case-finding recorded** Ethnicity/language ▼7% of other ethnicity (not English) had records of case finding vs 18% White British, 10% Pakistani (English), 11% Other (English) | N/A | CASP checklist for cohort studies. No concerns. There was missing data from the primary care data set, the quantity of which is unknown. Sample from one UK city with high levels of deprivation so may not be generalisable to the whole of the UK. Only those with questionnaire data were included, multiparous women and women living in more deprived areas were less likely to have completed a questionnaire. |

Management

(*Continued*)

**Table 2.** (Continued)

| Study | Study analysis methods (period and group studied) | Summary of findings—effects (Q1) ▾evidence that advantaged group is favoured ▴evidence that disadvantaged group is favoured ▸ little evidence of a difference between advantaged & disadvantaged groups | Summary of findings— explanation of effects (Q2) | Quality Appraisal |
|---|---|---|---|---|
| Redshaw & Henderson (2016) [38] | Secondary data analysis of the national maternity survey, 'Safely delivered' (AN, PN; mothers) | **AN Offered treatment for those who disclosed MH problems** **Education ▸, SES ▸, partnership status ▸** parity ▸ **Age**▾Bivariate NS. Multivariable: less likely age 16–19 (OR 0.27; 0.10, 0.78) and 40+ (0.34; 0.14, 0.83) (ref 30–34) **Ethnicity**▾Bivariate: less likely Asian (20%) and Black (18%) vs White 41%. Multivariable: Asian less likely (0.29; 0.18, 0.47). **AN Received support for those who disclosed MH problems** **Age ▸, parity ▸, partnership status ▸** **Ethnicity**▾ Bivariate: Asian less likely (53% vs 74% White). Multivariable: Asian less likely (0.35; 0.15, 0.78). **Education** ▾Bivariate: left school < age 17 less likely (58% vs 19+ 75%). Multivariable: left school < age 17 less likely (OR 0.52; 0.28, 0.97) **SES**▾ Bivariate: most deprived quintile 60% vs least 81%. Multivariable: NS. **AN Received advice for those who disclosed MH problems** **Education ▸, SES ▸, partnership status ▸, parity ▸, age ▸** **Ethnicity**▾ Bivariate: NS. Multivariable: Asian less likely (0.38; 0.17, 0.85) ref White **AN Received treatment for those who disclosed MH problems** **Ethnicity ▸, education ▸, SES ▸, partnership status ▸, parity ▸, age**▸ **PN Received support** **Education ▸, SES ▸, partnership status ▸, parity ▸, age**▸ **Ethnicity**▾ Bivariate: less likely Asian 43% and Black 46% vs White 67%. Multivariable: Asian less likely (0.37; 0.19, 0.71), Black NS. **PN Received advice** **SES ▸, partnership status ▸, parity ▸, age**▸ **Ethnicity**▾ Bivariate: Asian less likely 51% vs White 67%. Multivariable: Asian less likely (0.50; 0.27, 0.94) **Education** ▴Bivariate more likely left school < age 17 72% vs 19+ 62%. Multivariable: more likely left school < age 17 (1.75; 1.75, 3.12). **PN Received treatment** **SES ▸, partnership status ▸, parity ▸, age**▸ **Education** ▴ Bivariate: more likely left school < age 17 64% vs 19+ 42%. Multivariable: more likely left school < age 17 (2.26; 1.22, 4.18) **Ethnicity**▾Bivariate: less likely Asian 33% vs white 54%. Multivariable: Asian less likely (0.39; 0.18, 0.80) | N/A | As above |

(*Continued*)

**Table 2.** (Continued)

| Study | Study analysis methods (period and group studied) | Summary of findings—effects (Q1) ▼evidence that advantaged group is favoured ▲evidence that disadvantaged group is favoured ▶ little evidence of a difference between advantaged & disadvantaged groups | Summary of findings—explanation of effects (Q2) | Quality Appraisal |
|---|---|---|---|---|
| McCauley & Casson (2013) [36] | Qualitative, semi structured interviews (AN, PN; GPs) | **Education** ▼Management of women with lower levels of literacy (vs higher) **SES** ▼Management in areas of high deprivation (vs low) | **Education** ▼Low levels of literacy in areas of high deprivation are a barrier to comprehensive communication about treatment decisions **SES** ▼In highly deprived areas, some GPs take a more "paternalistic" approach to treatment decision making as they feel women do not want to be empowered to be involved in decision making | CASP checklist for qualitative studies. No concerns. The purposive sampling technique seemed inappropriate as practice managers could have acted as gatekeepers. Small sample size. Based in one Northern Ireland city with high deprivation so there may be limits in generalisability of the results. |
| Khan (2015) [34] and Russell & Lang (2013) [35] | Surveys, literature reviews, qualitative, interviews (AN, PN; Mothers, GPs, MW, HVs, other professionals) | **Place** ▼Management variation by where one lives **Ethnicity/language/culture** ▼ Management in consultations using an interpreter (vs not) and consultations with ethnic minority women (vs White) | **Place** Mothers and GPs acknowledged geographical differences in the quality and type of care received. **Ethnicity/language/culture** A third of GPs said cultural or translation factors could undermine the effectiveness of written resources used to support informed discussions. | As above |

GPs general practitioners, HVs health visitors, MWs midwives, RRR relative risk ratio, OR odds ratio, SES socio-economic status. The paper by Khan [34] was not included in NSC implementation review.

**Detection.** Five papers explored the detection of perinatal mental health problems (Table 2). The following ordering represents the organising categories in PROGRESS-Plus.

*Language, ethnicity and migration.* Five studies investigated or reported detection by ethnicity or language, no studies investigated detection by migration. Two of the studies had quality concerns; one over the under-representation of disadvantaged groups for one study [38] and uncertainty around sampling for the second [34, 35].

Differences in detection in relation to ethnicity or language were found in antenatal and postnatal studies. Comparing scores from the GHQ-28 in pregnancy to recording of information in GP data, Prady et. al. [22] found that women who were not White British were more than twice as likely to have potentially unidentified antenatal CMD, and less than half as likely to have a case-finding attempt recorded. The recording of the case finding attempt was lowest in women who spoke in their first language, which was not English. The majority of non-White British women in this study were of Pakistani origin. Redshaw and Henderson [38] found that Asian women, but not women of other ethnicity, were less likely to be asked about current, past or familial mental health problems compared to White women in the antenatal period. In this study, women were not asked to indicate whether they identified with being 'British', and were classified as White, Mixed, Black, Asian or other. In the postnatal period fewer women of any non-White ethnic group were asked about their mental health compared to White women.

Williams et.al [39] found midwives were less likely to ask depression case-finding questions with women who had limited English, and McGlone et. al. [37] reported concerns by midwives

that using an interpreter added a time burden to antenatal booking appointments which had implications for raising mental health conversations. Nearly half of midwives, health visitors and other non-GP health professionals felt that cultural factors and translation barriers affected case-finding [34, 35].

*Education*, *SES*, *Occupation*. Two studies, both quantitative, explored disparity by education, occupation or SES with quality concerns regarding under-representation of multiple, disadvantaged groups for one study [38].

Redshaw and Henderson [38] observed little variation by the age the mother left school and area-based deprivation (IMD) in relation to being asked about mental health in pregnancy, except women in the most deprived IMD quintile were *more* likely to be asked about family history. In the postnatal period however, women who had left education before the age of 19 and those living the most deprived quintile were less likely to be asked about their mental health. Prady et. al. [22] found that antenatal women of Pakistani origin, speaking English, who had a lower SES were more likely to have an unidentified CMD, compared to those with a higher SES. This relationship between detection and SES was not seen for women of other ethnic groups, and there was little variation by education attained. There was a relationship between *increased* detection for those who were previously employed versus currently employed but only in women who did not speak English.

*Personal characteristics associated with discrimination; parity, age, marital/partnership status*. Three studies explored disparity by parity, age and partnership, with quality concerns over under-representation of multiple, disadvantaged groups for one study [38].

*Parity*. Multiparous women were less likely to be asked about current or past mental health problems in the antenatal period in one study [39], in a second, there was little observed disparity [22]

*Age*. Antenatally there was a mixed picture for the relationship between age and factors relating to detection with Williams et. al. [39], finding that case-finding and detection was reduced for younger women, Prady et. al. [22] finding this relationship in White British women only and Redshaw and Henderson [38] find it reduced among older women compared with women of average child-bearing age. Postnatally, in the one study that examined it, Redshaw and Henderson [38] found that women <25 were at risk of disparity.

*Partnership*. Antenatally, Redshaw and Henderson [38] found little relationship between partnership status and women being asked about current or past/family history of mental health problems. Postnatally, women who were single were less likely to be asked compared with partnered women. Prady et. al. [22] found that cohabiting White British women were more likely to have undetected CMD compared to married, but there was no observed disparity for single women or for partnership status of ethnic minority women.

Relationship status can influence the screening of mental health problems from both women and health professional's point of views. Some women in an interview study noted that the presence of a partner at the booking appointment might limit how much they disclose and midwives reported that they would be less likely to use case screening questions if a partner attended [39].

**Management.** Three studies explored or reported management of perinatal mental health by at least one determinant (Table 2), with concerns over sampling [34, 35] and sample representativeness [38] for two studies.

*Language, ethnicity and migration*. In women identified as having a CMD in the antenatal period, Asian and Black women were less likely to be offered treatment, and Asian women were less likely to receive support or advice, compared to White women, but there was little variation between groups for those receiving treatment [38]. Postnatally, similar patterns were found but Asian women were also less likely to have received treatment than White women. In

an interview study, a third of GPs reported that cultural or translation factors could undermine the effectiveness of written resources used to support informed treatment discussions [34, 35]. No studies examined migration.

*Education*, *SES*, *Occupation*. Disparities were noted by mention of geographical differences in relation to the quality and type of care received [34, 35] and in the adoption of a treatment style with less empowerment and less shared decision making in areas of higher deprivation where lower literacy was more common [36]. In their survey of perinatal women, Redshaw and Henderson [38] found mixed effects antenatally for education and SES, with few observed effects for being offered treatment, receiving treatment or receiving advice but those who left school before 17 and in the most deprived IMD quintile were less likely to receive support. Postnatally women who left school before 17 were *more* likely to receive advice and treatment than those with more education; there were few observed differences by SES.

There were no studies examining occupation as a determinant of treatment.

*Personal characteristics associated with discrimination; parity*, *age*, *marital/partnership status*. There were few observed differences in management by age, parity and partnership status for the one study that examined these determinants, except that women age 16–19 and those 40+ were less likely to be offered treatment antenatally compared to women 30–34 [38].

**Absolute and relative differences.** In this section we present our calculations of standardised absolute prevalence estimates for overall detection and management, and absolute and relative estimates by determinants.

*Detection*. Results are presented in S1 Table. In Redshaw and Henderson [38], which has some concerns over disadvantaged, younger and single, ethnic minority and migrant women being less represented, the overall level of recollection by new mothers of being asked about current and previous problems antenatally and postnatally was relatively high overall; between 82.0% and 90.1%. Antenatally the largest disparity was between ethnic groups; with 6.9% fewer Asian women recollecting being asked about their mental health compared to White women, and 5.8% fewer Asian and 7.6% fewer women of other ethnicity asked about own or family mental health history. Postnatally some larger differences were apparent, particularly by ethnicity, with all minority groups disadvantaged; ranging from 22.3% fewer women of other ethnicity to 6.9% fewer women of mixed ethnicity women asked relative to White women. Women in the most deprived quintile were also less likely to be asked (8.4% fewer asked relative to those in the least deprived quintile). There was only small variation by parity, with 2–3% fewer multiparous asked relative to primaparous, and by education (4% fewer women with less education).

The level of antenatal case-finding activity recorded in GP notes reported in Prady et. al. [22] (no concerns) was very low at only 1.7%, and variation was also low, ranging between 1.4% fewer ethnic minority women preferring to use English to 2.1% fewer women who did not use English compared to White British women. In the postnatal period, with a higher prevalence of recorded case-finding activity overall (12.7%), variation was higher, with all minority groups having less case-finding activity recorded than White British women (range 7.4 to 11.4% lower). There were large differences in potentially missed CMD for all ethnic minority women (estimated between 37.7 to 43.0%), with relative risks over 2 for all groups, relative to White women. Data by determinants other than ethnicity were not reported for these measures.

*Management*. Results are presented in S2 Table. Only Redshaw and Henderson [38] reported quantitative data on management for women self-reporting mental health problems. There were some concerns about the representativeness of the sample analysed, and findings were not adjusted for severity of need. Our analysis of these data were hampered by small sample sizes in some categories, particularly in some ethnic minority groups. Disparities were

noted for some ethnic minority groups in women recalling being offered treatment antenatally, with 21.6% fewer Asian women and 23.3% fewer Black women having an offer compared to White women. There was little variation by SES or parity, but women with less education may have been more likely to have had a treatment offer. The proportion of women offered treatment antenatally in the whole sample was approximately 36%. Overall antenatal receipt of support, advice and treatment was approximately 70%, 72% and 46% respectively. Again there were ethnic group disparities, and some indication by lower SES. Postnatally, approximately 63% of women overall reported receiving support, 64% advice and 50% treatment. A similar pattern of disparity was noted as for antenatal receipt, with Asian women less likely to receive support (24.5% fewer), advice (16.5%) or treatment (20.5%) compared to White women. Women in the most disadvantaged SES group were also more likely to not receive postnatal help, but women with less education may have been more likely to receive treatment.

*Question 2. If implementation is not equitable, what are the potential reasons for this?*

Four studies reported on potential reasons for implementation of UK guidance not being equitable. There were concerns over the representativeness of the sample for one study [34, 35]. We found two main themes: Communication between health professionals and patients, and Provider perception of patient understanding.

**Communication between health professionals and patients.** Language was a dominant theme throughout all of the studies. The main way in which language was seen to be a barrier to equitable implementation was for mothers for whom English was not a first language. Some midwives reported that they would hesitate to use depression case finding questions, or omit them all together, if a woman's English "was not very good" [39 p.42].

The use of interpreters, and translated materials, was felt to be problematic. In a survey of non-GP health professionals nearly half felt that cultural or translation factors hindered starting a discussion about mental health [34, 35]. The use of an interpreter was considered to be time consuming, with the implication that this additional pressure acts as a barrier to identification [37]. Problems may be due in part to uncertainties around meaning. Around a third of GPs in a survey felt that translated materials, or other cultural factors, hindered informed decision-making around management [34, 35] and concerns over how the language of mental health problems translates to different cultures inhibited the use of screening questions by midwives [39].

Communication and information giving around management may be compromised for English speaking women with less literacy in that GPs prefer to communicate verbally [36].

**Provider perception of patient understanding.** GPs who worked with women from a deprived area with low literacy levels were described as taking a more "paternalistic" approach in deciding what and how much information to share with women if they felt the woman wanted to be less involved in the decision making [36]. However what the GPs deemed the best route (e.g. taking a paternalistic approach) was a judgement about what they thought a particular woman wanted. As noted in the Communication theme, uncertainty around understanding pervaded midwives decisions to use screening tools [39].

Not attributable to a theme was the finding that both women and GPs noted that geography played a part in both the type and quality of perinatal mental health care received [34, 35].

## Discussion

### Main findings

After re-analysing data underpinning national screening recommendations, we found evidence of inequitable implementation of the current UK guidance for the detection and management of mental health problems in perinatal women. The most consistent evidence of

disparity was for ethnic minority women, both for those who speak English, and those who do not. There was less consistent evidence for inequality of implementation for other axes of social disparity and for characteristics such as age, parity and partnership status. Explanations centred on translation and interpretation hindering communication, and hesitancy related to uncertainty from healthcare providers about what different cultures understand by mental health problems.

## Findings in context

Three of the studies in our review included samples that may not have been representative of the full spectrum of disadvantage seen in the UK, meaning that equity effects may have been under-estimated. However, our findings broadly accord with other UK literature in perinatal mental health (that are not focused on implementation of the current strategy) around concerns about translation accuracy and translators making cultural interpretations [40], and professional's perception that Black Caribbean mothers respond to mental health problems differently due to cultural identity [41]. Ethnic minority women may be less likely to receive treatments for common mental health problems [23] and treatment disparities for Black people have been noted in the general population [42]. Precision of estimates have also been hampered by small sample sizes in this literature [23, 42]. An experimental study, not perinatal specific, found that White GPs have difficulty diagnosing anxiety in ethnic minority patients [43]. An older identification study in primary care similarly found few disparities along axes of social disadvantage other than ethnic minority status [44].

## Limitations

**Data in our review.** Most of the studies where we found indications of potentially inequitable identification did not set out to explore disparities in care. The sampling, focus and interpretation of inequality in these studies may not have been explored systematically or adequately, and for this reason this does not comprise a definitive evidence base. The data on inequalities in management are drawn from selected samples; for example if Asian women are less likely to have their mental health problem detected, then fewer of them proceed to the management stage. Where there is disparity, the absolute and relative differences we have calculated, therefore, are likely to under-represent the population in need. There was under-representation of several axes of disadvantage in the two studies that provided quantitative data, which may have resulted in an under-estimate of disparity. Due to small samples and variation in concepts measured we were unable to determine whether there is a difference in the magnitude of inequality between the prenatal and postnatal periods.

**Our review.** We examined studies in an existing review after ascertaining that the methods used to identify relevant literature met our needs. In doing so, we may have missed some studies, including studies published after the NSC review, although we have discussed our findings in the context of this literature, and they accord. We did not study all axes of disadvantage, for example gender identity or sexual orientation, although we note that none of the studies we evaluated contained this information.

## Implications for policy and practice

This review indicates the presence of implementation inequalities for women who are not White British that is actionable and our findings suggest that a potential mismatch in cultural understanding is generated between interpreters and professionals. None of the studies in our review examined the effect of training on culturally effective methods to improve identification or management of perinatal mental health in ethnic minority women, but the uncertainty

observed here and in other studies suggests a knowledge deficit [40, 41, 45]. A lack of representation of ethnic minority perinatal healthcare professionals in the workforce may also be a contributing factor to less than optimal cultural competence of services [41]. Concerns around the additional time and complexity involved when communicating through an interpreter may suggest that longer appointments are warranted, which have workforce planning and cost implications.

The implications for practice and services centre on the fact that if a non-biased cohort of women with perinatal mental health problems are not identified, then needs assessments for services including treatment provision are incorrectly informed. These rates are needed in order to develop effective, targeted services and to reduce the potential negative outcomes for children. Disparities in service provision are not factored into reviews of effectiveness, and cost effectiveness, of screening programmes [46] or treatment programmes [47]. Due to the disparities seen in our review, we recommend that equity considerations are routinely included into evidence synthesis underpinning service needs planning, and into policy decisions such as universal screening. To not offer the same identification opportunities and appropriate services to all is discriminatory.

Our review examines disparities at the interface of the healthcare system, a downstream determinant in a conceptual framework of the generation of health inequalities [48]. While actions at this level will have little impact on the political and social structures that generate social hierarchy, they may help prevent the generation of further inequalities and we recommend that action is taken to mitigate future effects on already disadvantaged families. We focused on research conducted in the UK, but detection and treatment disparities by ethnicity, age, parity and SES have been reported in other countries that recommend universal perinatal screening such as the US and Australia [49–51], are likely ubiquitous, and require similarly urgent attention.

### Future research agenda

Our review indicates that there is a disparity for ethnic minority women, but has also highlighted many research gaps in this area needed to investigate causes and potential remediation strategies. Further quantitative analysis on large unselected samples with robust indication for need is warranted to improve the precision of estimates, and illuminate whether there are disparities for women in smaller minority groups. Studies in our review did not categorise White non-British women, although this group was thought to comprise around six per cent of the UK population in 2016 [52]. Little is known about what might comprise culturally effective care for minority White groups [53] and this needs further investigating. Similarly, although there is some overlap with English language ability, no studies in our review focused on effects in migrant women, who are especially at risk for poor mental health and underservice mediated by traumatic histories, stress, lack of support, fewer resources and unfamiliarity with care systems [54]. We understand little about intersectionality; for example cumulative or synergistic disparity effects in women who are minority ethnic, recently migrant, young, and multiparous. Latent class analysis of unselected samples may be useful method to categorise risk, with consideration of the complex environmental exposures that cause syndemic effects [55]. While larger samples might be generated using routinely collected general practice data, caution must be applied to ensure that indication for need has not been compromised by inequitable recording.

We are mindful that the reasons for inequitable identification and management will, at least in part, be related to variation by social group in the consequences for poor mental health, screening measure test performance and effectiveness of treatment, of which little is known

[45, 56]. Further inequalities research should attempt to incorporate such factors to better understand causes and consequences across the patient journey.

Our findings highlight a need for research into the use of interpreting services for mental health conversations. We need to understand how and why uncertainties are generated in the course of an interpreted consultation, and which uncertainties have basis, as a first step to designing remediation strategies. The views of interpreters, professionals and women should be triangulated, as interpreters find mental health consultations difficult on many levels [57]. As an external service, interpretation may seem an obvious candidate for contributing to inequalities, but this needs verifying using methods that observe and draw opinion from a range of actors about the functioning of a whole service, and that focus on disparities.

Women's voices were under-represented in our review, and their perspective is needed in order to explore how the suspected mismatch in perceptions related to mental health preferences and understanding of mental health problems might be generated. When asked, ethnic minority women say they want to be assessed for mental health problems perinatally [40], and a more nuanced understanding is needed of how this might best be carried out.

The possibility of inequalities generated through low literacy was raised in our findings, and we suggest that literacy is added to the–Plus categories of PROGRESS-Plus for future research in perinatal women. A focus on the reading level of written materials, found to be high on websites [58], and the effective translation of materials for cross-cultural understanding is warranted.

While there is some indication that training in culturally competent care may be effective in improving knowledge, attitudes, competence and patient satisfaction, there is a knowledge gap in effect on outcomes and in perinatal mental health [59, 60]. The effect of focus, orientation and content of UK culturally competent care training in effective delivery and outcomes is severely under-researched [53]. Delivered programs in mental health services do not focus on addressing the actual disparities experienced by ethnic minorities, such as diagnosis [61]. The effectiveness of increased appointment times for ethnic minority women on the detection and management of perinatal mental health problems should be assessed.

The design and evaluation of interventions should adhere to good practice in applied inequalities research and closely monitor effects on disadvantaged groups using structured tools to prevent inadvertent intervention-generated-inequalities [62]. Finally, a similar equity review incorporating evidence from across the world is warranted to estimate global burden, and potentially examine the distribution of inequalities by presence or absence of universal screening programmes.

## Conclusion

The identification and management of perinatal mental health problems is likely to be inequitable for ethnic minority women, and solutions are urgently needed to remediate it. Further quantitative and qualitative research should focus on clarifying whether other groups of women are at risk for inequalities, understand how mismatches in perception are generated and consider the systems in which this healthcare is delivered. Inequalities should be taken into account when considering evidence that underpins service needs planning and policy decision-making.

## Supporting information

**S1 Checklist. PRISMA-E 2012 checklist.**
(DOCX)

**S1 Table. Absolute and relative differences in case-finding equity.**
(DOCX)

**S2 Table. Absolute and relative differences in management equity (Redshaw and Henderson, 2016) [38].**
(DOCX)

## Author Contributions

**Conceptualization:** Stephanie L. Prady, Josie Dickerson, Tracey J. Bywater, Sarah L. Blower.

**Formal analysis:** Stephanie L. Prady, Charlotte Endacott.

**Funding acquisition:** Stephanie L. Prady.

**Investigation:** Stephanie L. Prady, Charlotte Endacott, Sarah L. Blower.

**Methodology:** Stephanie L. Prady.

**Project administration:** Stephanie L. Prady, Charlotte Endacott.

**Validation:** Stephanie L. Prady, Charlotte Endacott, Sarah L. Blower.

**Visualization:** Stephanie L. Prady, Sarah L. Blower.

**Writing – original draft:** Stephanie L. Prady, Charlotte Endacott.

**Writing – review & editing:** Stephanie L. Prady, Josie Dickerson, Tracey J. Bywater, Sarah L. Blower.

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
