## [Decision Letter · Decision Letter 0]

8 Dec 2020

PONE-D-20-31531

Inequalities in the identification and management of common mental disorders in the perinatal period: an equity focused re-analysis of a systematic review

PLOS ONE

Dear Dr. Prady,

Thank you for submitting your manuscript to PLOS ONE. After careful consideration, we feel that it has merit but does not fully meet PLOS ONE’s publication criteria as it currently stands. Therefore, we invite you to submit a revised version of the manuscript that addresses the points raised during the review process.

This article applies a unique lens to review  the published studies on identification and management of common mental disorders in the perinatal period.  It potentially adds great value to the existing literature.  As the reviewers noted, please be sure to explicitly address definitions that are being use throughout the paper. 

We look forward to receiving your revised manuscript.

Kind regards,

Sze Yan Liu, PhD

Academic Editor

PLOS ONE

Journal Requirements:

2. Please include a copy of Table 5 which you refer to in your text on page 21.

Reviewers' comments:

Reviewer's Responses to Questions

**Comments to the Author**

1. Is the manuscript technically sound, and do the data support the conclusions?

Reviewer #1: Yes

Reviewer #2: Yes

2. Has the statistical analysis been performed appropriately and rigorously? 

Reviewer #1: N/A

Reviewer #2: N/A

3. Have the authors made all data underlying the findings in their manuscript fully available?

Reviewer #1: Yes

Reviewer #2: Yes

4. Is the manuscript presented in an intelligible fashion and written in standard English?

Reviewer #1: Yes

Reviewer #2: Yes

5. Review Comments to the Author

Reviewer #1: This is an innovative review that addresses an important question. While it is specific to the UK, I do think that there are implications for other jurisdictions as well. I have made some suggestions below.

Definition of the equity variables:

->In the methods equity variables should be more clearly defined. There is some discussion of this in the analysis section as it pertains to the quantitative papers, but should be expanded and include how they were defined for all papers.

->Why was parity considered as an equity variable?

->What groups were a priori considered advantaged vs. disadvantaged (as per Table 2)?

->For ethnicity, did you consider ethnicity and race (some of the categories seem to suggest so)?

Table 1 – Within column “Reported results by equity” consider defining how each of these variables was measured/defined in each study. E.g. self-reported ethnicity with XX categories.

Table 2

->The presentation of the results in the column “Summary of findings – effects (Q1)” is somewhat confusing. The referent group is not always clear. Sometimes the use of shorthand is a little too liberal. For example, for Khan 2015 and Russell/Lang 2013, not clear what “Ethnicity/language▼ Consultations using an interpreter, consultations with ethnic minority women” means.

Presentation of results generally

-> Headings should be consistently labeled (italics vs bolding etc) for clarity.

->“Absolute and Relative differences” are presented separately from the results above it – it might be better to integrate?

Other comments:

->“Paternalistic” is used in several places. Was this your interpretation that is was paternalistic, or labelled paternalistic in the study? If the latter, was this by the providers/patients or the study authors?

->How do your results fit into literature from other jurisdictions? How could lessons from your study inform practices in other countries?

->Why was only one of the Prady papers (ref 22) included and not the other (ref 23)?

Reviewer #2: Thank you for this well-written and insightful paper. The authors have provided a unique contribution to the literature by applying a health equity lens to their re-review of UK-based studies examining detection and management of CMDs in the perinatal populations. A number of important barriers to equitable implementation of population-based programs are identified and well-considered suggestions for addressing these inequalities from research and practice perspectives are provided. Findings have applicability outside the UK.

The Methods and Results were clearly presented and led logically to the Discussion and Conclusion. The limitations of the study were also adequately addressed.

Minor suggestions for improvement:

1. Additional context around why migration (and also years since migration) specifically is an important factor to consider would be helpful

2. Possible reasons for the observed differences in the magnitude of inequality among pregnant vs postnatal women would also be of interest

3. Minor amendment: line 76, while the PHQ-2 and Whooley questions are similar they are not the same (for example, they ask about different intervals of time – PHQ-2 – last 2 weeks; Whooley questions – last month)

I thank the authors again for this important work.

6. PLOS authors have the option to publish the peer review history of their article (what does this mean?). If published, this will include your full peer review and any attached files.

Reviewer #1: No

Reviewer #2: No

---

## [Author Response · Author response to Decision Letter 0]

8 Jan 2021

We thank the reviewers and editor for the opportunity to revise our paper and respond to each point below. 

Journal Requirements:

* Thank you. We have restructured the title page and headings in the abstract (not track changes). 

2. Please include a copy of Table 5 which you refer to in your text on page 21.

* Apologies, this was an error. The information is to be found in Supplementary Material Table 1, which is signposted. We have removed the words ‘(Table 5)’

*We have split the supplementary file into S1 Table and S2 Table, renamed accordingly, and added captions at the end of the manuscript.

5. Review Comments to the Author

Reviewer #1: This is an innovative review that addresses an important question. While it is specific to the UK, I do think that there are implications for other jurisdictions as well. I have made some suggestions below.

* Thank you for your review. 

Definition of the equity variables:

->In the methods equity variables should be more clearly defined. There is some discussion of this in the analysis section as it pertains to the quantitative papers, but should be expanded and include how they were defined for all papers.

->Why was parity considered as an equity variable?

->What groups were a priori considered advantaged vs. disadvantaged (as per Table 2)?

->For ethnicity, did you consider ethnicity and race (some of the categories seem to suggest so)?

* We agree that our treatment of this was too brief and have added a new section, Categorisation of determinants, to the Methods (line 159), which details the distinction of advantaged from disadvantaged and provides information on why age and parity were included. The framework we have applied (PROGRESS-Plus) uses the word ‘race’ in the classification group that also includes ‘ethnicity’, ‘language’ and ‘culture’, and hence this is why the word appears in the paper. In the UK, self-ascribed ‘ethnicity’ and not ‘race’ is used in official statistics, including the census, which is the (rough) basis for measurement in research. By default, then, we were considering ‘ethnicity’ and not ‘race’, and have not used the word ‘race’ in the manuscript further than the definition. We think that the new section makes it more clear that the four concepts are simply grouped within this section as a whole, and it was not our intention to analyse by ‘race’. It remains a difficulty to remove it altogether, as the R in race forms part of the acronym PROGRESS (the framework is not UK-specific). 

Table 1 – Within column “Reported results by equity” consider defining how each of these variables was measured/defined in each study. E.g. self-reported ethnicity with XX categories.

* Thank you, that is a logical progression from being more clear about classification in the Methods, and we have made these suggested changes in Table 1. 

Table 2

->The presentation of the results in the column “Summary of findings – effects (Q1)” is somewhat confusing. The referent group is not always clear. Sometimes the use of shorthand is a little too liberal. For example, for Khan 2015 and Russell/Lang 2013, not clear what “Ethnicity/language▼ Consultations using an interpreter, consultations with ethnic minority women” means.

* Maintaining legibility while presenting this amount of complex information in Table 2 has been challenging. We are unable to see how we could completely overhaul it, so we have made the following changes which we think improves its readability. (1) We have moved the legend for ▲▼► to the column heading. (2) We have been more liberal with the descriptions for qualitative papers where the classification between advantaged and disadvantaged is not defined, and often in context to the interviewees’ narrative. 

Presentation of results generally

-> Headings should be consistently labeled (italics vs bolding etc) for clarity.

* Thank you, we have corrected the erroneous heading levels. 

->“Absolute and Relative differences” are presented separately from the results above it – it might be better to integrate?

* We did consider this when compiling the results, but we felt it important to keep the two separate; the former being results reported as per each paper, the latter being standardised calculations that we have performed. We have added a signpost under the heading ‘Absolute and relative differences’ (line 343) reminding the reader what this section represents. 

Other comments:

->“Paternalistic” is used in several places. Was this your interpretation that is was paternalistic, or labelled paternalistic in the study? If the latter, was this by the providers/patients or the study authors?

* This was a descriptor used by the study authors to describe the approach taken by GPs for some women who seem reluctant to take ownership of their health. We agree that we have overused the word and not made its origins clear. To rectify, we have used quotes around it in Table 2, and also noted that it was described in the paper as such (line 413), and used other descriptors to describe the management approach taken by GPs (line 327). 

->How do your results fit into literature from other jurisdictions? How could lessons from your study inform practices in other countries?

* We have added a sentence to ‘Implications for Policy and Practice’ in the Discussion (line 491) that outlines similar patterns in some other countries. We have also added that future research should include an international equity review (line 548). 

->Why was only one of the Prady papers (ref 22) included and not the other (ref 23)?

* We included papers that were included in the NSC review. Reference 23 was excluded from the NSC review because “Women selected were diagnosed with mental health disorder prior to becoming pregnant” (Page 105, NSC review), i.e. the analysis was of an already selected sample.

Reviewer #2: Thank you for this well-written and insightful paper. The authors have provided a unique contribution to the literature by applying a health equity lens to their re-review of UK-based studies examining detection and management of CMDs in the perinatal populations. A number of important barriers to equitable implementation of population-based programs are identified and well-considered suggestions for addressing these inequalities from research and practice perspectives are provided. Findings have applicability outside the UK.

* Thank you for your review. 

The Methods and Results were clearly presented and led logically to the Discussion and Conclusion. The limitations of the study were also adequately addressed.

Minor suggestions for improvement:

1. Additional context around why migration (and also years since migration) specifically is an important factor to consider would be helpful

* Thank you, it was our omission that we had not drawn this out. We have added a sentence in the discussion (line 502) outlining the consideration and highlighting the lack of evidence.

2. Possible reasons for the observed differences in the magnitude of inequality among pregnant vs postnatal women would also be of interest

* Due to the variation in the concepts measured between pregnant and postnatal women across (and within) studies, and the generally small sample sizes, we are unclear whether there are differences between the periods, and have added this as a limitation of the data in the discussion (line 458). 

3. Minor amendment: line 76, while the PHQ-2 and Whooley questions are similar they are not the same (for example, they ask about different intervals of time – PHQ-2 – last 2 weeks; Whooley questions – last month)

* We have corrected this and cited Whooley et al (1997) paper in place of the PHQ-2 reference (line 76). 

I thank the authors again for this important work.

---

## [Decision Letter · Decision Letter 1]

11 Feb 2021

PONE-D-20-31531R1

Inequalities in the identification and management of common mental disorders in the perinatal period: an equity focused re-analysis of a systematic review

PLOS ONE

Dear Dr. Prady,

Thank you for submitting your manuscript to PLOS ONE. After careful consideration, we feel that it has merit but does not fully meet PLOS ONE’s publication criteria as it currently stands. Therefore, we invite you to submit a revised version of the manuscript that addresses the points raised during the review process.

The new additions to your manuscript were very helpful in understanding the framework you are applying.  One reviewer had additional comments along those lines which I think would help further clarify your conceptual approach.

We look forward to receiving your revised manuscript.

Kind regards,

Sze Yan Liu, PhD

Academic Editor

PLOS ONE

Reviewers' comments:

Reviewer's Responses to Questions

**Comments to the Author**

1. If the authors have adequately addressed your comments raised in a previous round of review and you feel that this manuscript is now acceptable for publication, you may indicate that here to bypass the “Comments to the Author” section, enter your conflict of interest statement in the “Confidential to Editor” section, and submit your "Accept" recommendation.

Reviewer #1: (No Response)

2. Is the manuscript technically sound, and do the data support the conclusions?

Reviewer #1: Yes

3. Has the statistical analysis been performed appropriately and rigorously? 

Reviewer #1: Yes

4. Have the authors made all data underlying the findings in their manuscript fully available?

Reviewer #1: Yes

5. Is the manuscript presented in an intelligible fashion and written in standard English?

Reviewer #1: Yes

6. Review Comments to the Author

Reviewer #1: Thank you for addressing my comments. The edited sections are much clearer. Overall I think this is an interesting paper and addresses an important topic. I have just two remaining comments:

1) My main remaining question is regarding the rationale to include parity as an equity variable as it is more atypical, and age is already included which is the rationale given. Could you perhaps expand on your rationale or reference some prior work where this has been included as an equity variable? There are certainly ways parity I image parity could impact service use (e.g. multiparity making it harder to attend appointments if no childcare provided) but the rationale should just be more explicitly stated and referenced.

2) A limitation regarding the axes of inequity that were not under study (e.g. gender identity, sexual orientation, etc) could be considered.

7. PLOS authors have the option to publish the peer review history of their article (what does this mean?). If published, this will include your full peer review and any attached files.

Reviewer #1: No

---

## [Author Response · Author response to Decision Letter 1]

15 Feb 2021

Reviewer #1: Thank you for addressing my comments. The edited sections are much clearer. Overall I think this is an interesting paper and addresses an important topic. I have just two remaining comments:

1) My main remaining question is regarding the rationale to include parity as an equity variable as it is more atypical, and age is already included which is the rationale given. Could you perhaps expand on your rationale or reference some prior work where this has been included as an equity variable? There are certainly ways parity I image parity could impact service use (e.g. multiparity making it harder to attend appointments if no childcare provided) but the rationale should just be more explicitly stated and referenced.

2) A limitation regarding the axes of inequity that were not under study (e.g. gender identity, sexual orientation, etc) could be considered.

Thank you for your review, we are very grateful for your suggestions as the changes we have made have considerably strengthened our paper. Below we explain how we have addressed your remaining comments. 

1) We agree it was not clear why we thought parity should be investigated, as what we had written equated it with age. We have amended this and added a sentence (page 8, line 168) to make our rationale clear about why we thought increased parity might be a disadvantaging factor in the UK context. We have added a notation in the Discussion indicating that inequity with increased parity has been noted in a non-UK study (page 29, line 494). 

2). We have added a limitation relating to ‘our review’ (page 28 line 466) around the axes of inequity we did not apply as suggested.

---

## [Decision Letter · Decision Letter 2]

3 Mar 2021

Inequalities in the identification and management of common mental disorders in the perinatal period: an equity focused re-analysis of a systematic review

PONE-D-20-31531R2

Dear Dr. Prady,

We’re pleased to inform you that your manuscript has been judged scientifically suitable for publication and will be formally accepted for publication once it meets all outstanding technical requirements.

Kind regards,

Sze Yan Liu, PhD

Academic Editor

PLOS ONE

Additional Editor Comments (optional):

Reviewers' comments:

Reviewer's Responses to Questions

**Comments to the Author**

1. If the authors have adequately addressed your comments raised in a previous round of review and you feel that this manuscript is now acceptable for publication, you may indicate that here to bypass the “Comments to the Author” section, enter your conflict of interest statement in the “Confidential to Editor” section, and submit your "Accept" recommendation.

Reviewer #1: All comments have been addressed

2. Is the manuscript technically sound, and do the data support the conclusions?

Reviewer #1: (No Response)

3. Has the statistical analysis been performed appropriately and rigorously? 

Reviewer #1: (No Response)

4. Have the authors made all data underlying the findings in their manuscript fully available?

Reviewer #1: (No Response)

5. Is the manuscript presented in an intelligible fashion and written in standard English?

Reviewer #1: (No Response)

6. Review Comments to the Author

Reviewer #1: (No Response)

7. PLOS authors have the option to publish the peer review history of their article (what does this mean?). If published, this will include your full peer review and any attached files.

Reviewer #1: No

---

## [Editor Report · Acceptance letter]

4 Mar 2021

PONE-D-20-31531R2 

Inequalities in the identification and management of common mental disorders in the perinatal period: an equity focused re-analysis of a systematic review 

Dear Dr. Prady:

I'm pleased to inform you that your manuscript has been deemed suitable for publication in PLOS ONE. Congratulations! Your manuscript is now with our production department. 

Kind regards, 

on behalf of

Dr. Sze Yan Liu 

Academic Editor

PLOS ONE